# Comparative Study of Several Fe Deficiency Responses in the *Arabidopsis thaliana* Ethylene Insensitive Mutants *ein2-1* and *ein2-5*

**DOI:** 10.3390/plants10020262

**Published:** 2021-01-29

**Authors:** Macarena Angulo, María José García, Esteban Alcántara, Rafael Pérez-Vicente, Francisco Javier Romera

**Affiliations:** 1Department of Agronomy, Campus de Excelencia Internacional Agroalimentario CeiA3 de Rabanales, Universidad de Córdoba, Edificio Celestino Mutis, 14071 Córdoba, Spain; b02anpem@uco.es (M.A.); ag1alvae@uco.es (E.A.); ag1roruf@uco.es (F.J.R.); 2Department of Botany, Ecology and Plant Physiology, Campus de Excelencia Internacional Agroalimentario CeiA3 de Rabanales, Universidad de Córdoba, Edificio Celestino Mutis, 14071 Córdoba, Spain; bv1pevir@uco.es

**Keywords:** ferric reductase activity, ethylene, iron, nitric oxide, root hairs, signaling, *S*-nitrosoglutathione

## Abstract

Iron (Fe) is an essential micronutrient for plants since it participates in essential processes such as photosynthesis, respiration and nitrogen assimilation. Fe is an abundant element in most soils, but its availability for plants is low, especially in calcareous soils. Fe deficiency causes Fe chlorosis, which can affect the productivity of the affected crops. Plants favor Fe acquisition by developing morphological and physiological responses in their roots. Ethylene (ET) and nitric oxide (NO) have been involved in the induction of Fe deficiency responses in dicot (Strategy I) plants, such as Arabidopsis. In this work, we have conducted a comparative study on the development of subapical root hairs, of the expression of the main Fe acquisition genes *FRO2* and *IRT1*, and of the master transcription factor *FIT*, in two *Arabidopsis thaliana* ET insensitive mutants, *ein2-1* and *ein2-5*, affected in EIN2, a critical component of the ET transduction pathway. The results obtained show that both mutants do not induce subapical root hairs either under Fe deficiency or upon treatments with the ET precursor 1-aminocyclopropane-1-carboxylate (ACC) and the NO donor *S*-nitrosoglutathione (GSNO). By contrast, both of them upregulate the Fe acquisition genes *FRO2* and *IRT1* (and *FIT*) under Fe deficiency. However, the upregulation was different when the mutants were exposed to ET [ACC and cobalt (Co), an ET synthesis inhibitor] and GSNO treatments. All these results clearly support the participation of ET and NO, through EIN2, in the regulation of subapical root hairs and Fe acquisition genes. The results will be discussed, taking into account the role of both ET and NO in the regulation of Fe deficiency responses.

## 1. Introduction

Iron (Fe) is very abundant in most soils, mainly as Fe^3+^, although its availability to plants is low, especially in calcareous soils [1,2,3]. On the other hand, excessive Fe accumulation by the plant may lead to toxic effects [4,5]. Therefore, Fe acquisition is highly regulated. Dicot (Strategy I) plants, such as Arabidopsis, need to reduce Fe^3+^ to Fe^2+^ by means of a plasma membrane ferric reductase, encoded by the *FRO2* gene, prior to its root absorption through a Fe^2+^ transporter, encoded by the *IRT1* gene [4,6,7]. When grown under Fe deficiency, dicot plants induce several physiological and morphological responses (mainly in their roots) aimed at facilitating the mobilization and acquisition of this nutrient [4,6,7]. Among the physiological responses, dicot plants enhance both ferric reductase activity (FRA; due to increased expression of *AtFRO2*-like genes) and Fe^2+^ uptake capacity (due to increased expression of *AtIRT1*-like genes) [4,7]. In addition to physiological responses, dicot plants can develop some morphological responses in their roots, such as subapical root hairs, root epidermal transfer cells and cluster roots, also named proteoid roots [7,8]. All these root modifications enhance Fe uptake by increasing the contact surface of roots with soil and by chemically modifying the soil environment [9].

The regulation of the Fe-related genes associated with physiological responses, such as *FRO2* and *IRT1*, is not totally understood, but in the last few years several transcription factors (TFs) that participate in their activation have been found. The master regulator of most Fe acquisition genes in *Arabidopsis* is the FIT (bHLH29) TF [4,6,7,10,11]. The Arabidopsis *fit* mutant is very chlorotic and lacks the ability to activate most Fe deficiency responses in roots [7]. In Arabidopsis, the FIT regulatory network comprises other bHLH TFs, such as bHLH38, bHLH39 and others [10,11]. Some of these TFs, such as bHLH38 and bHLH39, can interact with FIT to form heterodimers that activate the expression of the Fe acquisition genes *FRO2* and *IRT1* [4,10,11]. *FIT* is induced in roots in response to Fe deficiency, while other *bHLHs* are induced in both roots and leaves in response to Fe deficiency [4].

Once adequate Fe has been absorbed, Fe deficiency responses need to be switched off to minimize Fe toxicity and energy costs. Their regulation is not fully understood, but several hormones and signaling substances, such as auxin, ethylene (ET) and nitric oxide (NO), which increase their production in Fe-deficient roots, have been proposed to participate in the activation of most responses in dicot plants [4,7,12,13,14,15,16,17,18,19,20,21]. Auxin, ET and NO are closely interrelated in a complex manner since each one can affect the production and/or distribution of the other ones [19,22,23,24,25]. For example, it has been found that the expression of genes encoding enzymes implicated in ET synthesis, and ET itself, are enhanced by NO treatments, while NO accumulates in the subapical region of roots treated with the ET precursor 1-aminocyclopropane-1-carboxylic acid (ACC) [24,26]. Besides ET synthesis, Fe deficiency also affects ET responsiveness by altering the expression of genes implicated in ET signaling, such as *EIN2*, *EIN3*, *EILs* and *ERFs* [7,18,19]. *EIN2* is upregulated in Arabidopsis roots under Fe deficiency [27] and also in plants overexpressing the *PDF1.1* gene, which causes upregulation of Fe acquisition genes [28]. In contrast to auxin, ET and NO, other substances, such as cytokinins, jasmonic acid (JA) and brassinosteroids, have been involved in the suppression of Fe deficiency responses [4,7,16,19].

ET is synthesized from methionine via a pathway that requires several enzymes, such as ACC synthases and ACC oxidases, and where ACC is the immediate ET precursor [29]. ET synthesis can be blocked by using several ethylene inhibitors, such as cobalt (Co), or promoted by applying the ET precursor ACC [7,12,13]. Although ethylene’s mode of action is not fully understood, a linear canonical signaling pathway has been proposed in Arabidopsis [30,31,32]:ET ─╢ ET receptors → CTR1 ─╢ EIN2 → EIN3/EILs → ERFs → ET responses(1)

In this pathway, CTR1 is a kinase, EIN2 is a transmembrane protein located in the endoplasmic reticulum (ER) membrane, and EIN3, EILs and ERFs are TFs [30,31,32].

EIN2 possesses a Nramp-like transmembrane domain and a cytosolic **C**OOH **end** (CEND; Figure 1) domain [32,33,34,35]. The role of the Nramp transmembrane domain is not yet clear, while two roles have been proposed for the CEND portion [32,33,34,35]. The Nramp portion shows similarity to the Nramp family of metal-ion transporters [32,36]. However, no metal transport activity has been detected in heterologously expressed EIN2 [32], although its possible role as a sensor of divalent metals has been suggested [36]. In the absence of ET, CTR1 phosphorylates EIN2, preventing the cleavage and translocation of CEND into the nucleus. In the presence of ET, CTR1 is inactivated, resulting in dephosphorylation of EIN2 and its cleavage. CEND is then translocated into the nucleus, where it interacts with the EIN2 Nuclear Associated Protein 1 (ENAP1) and potentially Histone Acetyl Transferases (HATs), resulting in histone acetylation. This causes an uncompacting of chromatin, resulting in more EIN3/EIL1 binding to target genes and ultimately transcription activation (Figure 1) [32,35,37,38]. In addition to this role in the nucleus, CEND in the cytosol can block the mRNA translation of the F-box proteins EBF1 and EBF2, which participate in the proteasomal degradation of EIN3 and EIL1, thus promoting their accumulation [32,33,34]. EIN3 and EIL1 further regulate the expression of the ET response factor (ERF) family TFs [30,31,32,39]. The ERF TFs act downstream of EIN3, and EIL1, to activate or repress ET-responsive genes, although some ERFs can be activated by ET-independent TFs, not related to EIN3 [40]. In Arabidopsis, as in other plant species, many mutants altered in the different ET signaling genes have been identified. Among them, the *ein2* mutants are insensitive to most responses to ET [30,35,36]. More recently, the existence of an alternate “non canonical” route for ET signaling, besides the one including EIN2, is further supported by different experimental results (Figure 1; [7,30,32,41]). For example, the expression of several Fe acquisition genes in response to either ET precursors or inhibitors is altered in the Arabidopsis *ein2-1* mutant [27,42]. Moreover, Kim et al. [43] have shown that this mutant is capable of responding to ET if JA levels or signaling are low.

The implication of ET in the regulation of Fe deficiency responses in dicot plants is supported by many experimental results. For example, the addition of ET inhibitors, such as Co, to Fe-deficient plants inhibits the induction of most Fe deficiency responses, such as enhanced FRA and subapical root hairs, while the addition of ET or ACC to Fe-sufficient plants induces them [7,12,13,19,27,46,47]. Furthermore, ET also affects the expression of the genes controlling these responses [7,19,27,44,46,47,48]. For example, ET upregulates *FIT* expression and, consequently, the Fe acquisition genes *FRO2* and *IRT1*, activated by this key TF (Figure 1) [27,46,48]. The link between ET and FIT has been recently reinforced. It has been shown that EIN3 and EIL1, two TFs in the ET signaling pathway, interact with MED16 (Mediator) and MED25 to form a complex implicated in *FIT* transcription [4,44]. Moreover, EIN3 and EIL1 can also influence the posttranscriptional stability of FIT [48].

ET, along with auxin, NO, and other hormones and signaling substances, has also been implicated in the regulation of Fe deficiency morphological responses, such as subapical root hairs [7,8,13,14,19]. In addition to the results obtained with ET inhibitors and precursors, several ET mutants have been used to study the role of ET in the development of these morphological changes. In relation to the EIN2 protein, several mutants have been identified in *Arabidopsis*, such as *ein2-1* and *ein2-5* [33,36,43,49]. The *ein2-5* mutation is located in the Nramp transmembrane domain, while the *ein2-1* mutation is located in the CEND cytosolic portion (Figure 1) [36]. It has been shown that the *Arabidopsis* ET insensitive *ein2-1* mutant did not develop subapical root hairs either under Fe-deficiency or upon ET treatments, while the wild-type did [13,27]. Similarly, the *ein2-5* mutant did not develop subapical root hairs upon ACC treatment [49]. Results obtained with the *ein2-1* mutant also suggest that Fe deficiency physiological and morphological responses can be regulated through different ET signaling pathways. For instance, the development of subapical root hairs is impaired in the *ein2-1* mutant (see above), while enhanced FRA and the expression of Fe acquisition genes is not [13,27,42].

In previous works, it was found that the addition of Co, an ET synthesis inhibitor, at 50 µM final concentration, to Fe-deficient wild-type Columbia plants drastically inhibited several Fe deficiency responses, such as FRA enhancement and *FIT*, *FRO2* and *IRT1* upregulation [27,46]. However, the application of Co, at 50 µM final concentration, to Fe-deficient *ein2-5* plants inhibited FRA enhancement but did not inhibit *FIT*, *FRO2* or *IRT1* upregulation (unpublished results). The objective of this work was to look further into the above lack of inhibition of the *ein2-5* mutant by comparing it with the *ein2-1* mutant under different Co concentrations, and also under other treatments, such as ACC and *S*-nitrosoglutathione (GSNO) applications. GSNO is derived from glutathione (GSH) and NO and is regarded as an intracellular NO reservoir as well as a vehicle of NO throughout the cell [50]. Although endogenous GSNO, present at very low concentrations, does not behave exactly as NO [51,52], exogenous applied GSNO, at the concentrations used, is considered to be a NO donor [14,22,26].

## 2. Results

In wild-type Columbia plants, Fe deficiency induces a suite of genes in roots, including those involved in Fe acquisition, such as *FRO2*, *IRT1* and *FIT* [7,27,46]. In previous works, it was shown that the upregulated expression of the Fe deficiency induced genes *FRO2*, *IRT1* and *FIT* in wild-type Columbia plants was suppressed by ET inhibitors, such as cobalt (Co), applied at 50 µM final concentration [27,46]. However, preliminary experiments in our lab showed that the application of Co, at 50 µM final concentration, to Fe-deficient *ein2-5* plants did not inhibit *FIT*, *FRO2* or *IRT1* upregulation (unpublished results).

In this work, we wanted to verify the effect of Co on the expression of the Fe-related genes described above, and also on ferric reductase activity (FRA), in the Arabidopsis ET insensitive *ein2.5* mutant but also in the *ein2.1* mutant. In addition, we wanted to compare the ability of both mutants to respond to ET and NO treatments. As described below, the behavior of the mutants in response to Co (ET synthesis inhibitor), ACC (ET precursor) and GSNO (NO donor) was different in the induction of physiological responses.

### 2.1. Effect of Co on Ferric Reductase Activity (FRA) and FRO2, IRT1 and FIT Expression

*FRO2*, *IRT1* and *FIT* expression was clearly induced in both mutants, *ein2.1* and *ein2.5*, under Fe deficiency (Figure 2).

*IRT1* and *FIT* attained the highest values of their upregulation in the *ein2.1* mutant, while *FRO2* reached the highest values in the *ein2.5* mutant (Figure 2). The effect of Co (ET inhibitor) on the expression of the above genes in Fe-deficient plants was different depending on the mutant. While in the *ein2.1* mutant, Co drastically inhibited the expression of all the genes studied, in the *ein2.5* mutant, Co had no effect in any case except on *FIT* expression, where its relative expression decreased with the higher Co doses (75 and 100 µM; Figure 2).

FRA was greatly enhanced under Fe deficiency in both mutants and was drastically inhibited upon Co treatments in both mutants (Figure 3).

Collectively, the results show that the Fe deficiency physiological responses studied (FRA and *FRO2*, *IRT1* and *FIT* expression) are induced by Fe deficiency in both mutants. However, both mutants behave differently in response to the ET synthesis inhibitor Co: it drastically inhibits *FRO2* and *IRT1* expression in Fe-deficient *ein2-1* plants but not in Fe-deficient *ein2-5* plants.

### 2.2. Effect of ACC on Ferric Reductase Activity (FRA) and FRO2, IRT1 and FIT Expression

Previous results showed that the *ein2.1* mutant was able to upregulate *FRO2*, *IRT1* and *FIT* expression in response to ET treatments as the wild-type did [27]. The results obtained in this work agree with these previous ones, showing that these genes are upregulated by ACC (ET precursor) in both *ein2* mutants (Figure 4). However, while *FRO2* and *IRT1* were greatly induced by ACC in Fe-sufficient *ein2.1* mutant plants, mainly at 5 µM, the induction was much lower in Fe-sufficient *ein2.5* mutant plants (Figure 4). Even at ACC 5 µM, the *ein2.5* mutant did not reach the expression levels attained by the *ein2.1* mutant at ACC 1 µM (Figure 4). However, these differences were not remarkable in relation to *FIT* expression (Figure 4).

In relation to FRA, at ACC 1 µM, its enhancement was higher in Fe-sufficient *ein2.1* mutant plants than in Fe-sufficient *ein2-5* mutant plants (Figure 5). At ACC 5 µM, the enhancement was similar in both mutants (Figure 5).

Collectively, the results show that both mutants induce the Fe deficiency physiological responses studied (FRA and *FRO2*, *IRT1* and *FIT* expression) in response to ACC. However, the intensity of the induction was generally lower in the *ein2-5* mutant.

### 2.3. NO Accumulation in Wild-Type Columbia, ein2.1 and ein2.5 Roots in Response to Fe Deficiency and ACC Treatment

As shown in Figure 6, NO accumulated in wild-type Columbia roots under Fe deficiency and also in response to ACC treatments (Figure 6). In the *ein2.1* mutant, the results were similar to the wild-type, although they presented slightly lower NO accumulation, mainly upon ACC treatment (Figure 6). Finally, the *ein2.5* mutant was able to accumulate NO in response to Fe deficiency but not in response to ACC treatment (Figure 6).

Collectively, the results show that both mutants accumulate NO in the subapical region of the roots in response to Fe deficiency, as occurred in the wild-type Columbia. However, while the *ein2.1* mutant also accumulates NO in response to ACC treatment, as occurred in the wild-type Columbia, this accumulation does not occur in the *ein2-5* mutant.

### 2.4. Effect of GSNO on Ferric Reductase Activity (FRA) and FRO2, IRT1 and FIT Expression

Under Fe sufficient conditions, *FRO2*, *IRT1* and *FIT* expression was induced by GSNO treatment in the *ein2.1* mutant (Figure 7), as occurred in the wild-type Columbia [26]. However, in the *ein2.5* mutant, GSNO treatment did not induce the expression of any of these genes but inhibited it (Figure 7).

Under Fe-deficient conditions, the results were similar: *FRO2*, *IRT1* and *FIT* expression was induced by GSNO treatment in the *ein2.1* mutant but not in the *ein2.5* mutant (Figure 8). In the *ein2-1* mutant, *FRO2* and *IRT1* expression attained much higher values in the GSNO-treated plants under Fe deficiency (Figure 8) than under Fe sufficiency (Figure 7).

In relation to FRA, it was induced by the GSNO treatment in both mutants under Fe deficient conditions, with the enhancement being much higher in the *ein2.1* mutant (Figure 9).

Under Fe-sufficient conditions, there was no effect of GSNO on FRA in any of the mutants (data not shown).

Collectively, the results show that the *ein2-1* mutant induces all the Fe deficiency physiological responses studied (FRA and *FRO2*, *IRT1* and *FIT* expression) in response to GSNO, while the *ein2-5* mutant only slightly induced FRA but not *FRO2*, *IRT1* and *FIT* expression.

### 2.5. Effect of ACC and GSNO on the Development of Subapical Root Hairs

The addition of ACC (at 1 or 10 µM final concentration) or GSNO (at 100 or 500 µM final concentration) to Fe-sufficient *ein2-1* and *ein2-5* plants did not induce the development of subapical root hairs as occurred in the wild-type Columbia, even with lower ACC and GSNO doses (Figure 10). None of the mutants induced the development of subapical root hairs under Fe deficiency, while the wild-type did (results not shown).

## 3. Discussion

Ethylene (ET), along with other hormones and signaling substances, such as auxin and nitric oxide (NO), has been involved in the activation of physiological and morphological responses to Fe deficiency in dicot plants [7,12,13,14,15,17,18,19,20,23,26,27,46,53]. Fe deficiency affects ET synthesis but also ET responsiveness, related to changes in the expression of genes involved in ET signaling, such as *ETR1*, *CTR1*, *EIN2*, *EIN3*, *EILs* and *ERFs* [7,18,19,27]. In this work, we have studied the induction of some Fe deficiency physiological responses in two *Arabidopsis* ET signaling mutants: *ein2-1* and *ein2-5* [24,33,36,49]. The *ein2* mutants have been described as insensitive to most responses to ET [30,31,32,35,36]. In fact, EIN2 is considered one of the critical players in the linear signaling pathway proposed for ET action (see Introduction) [30,31,32,34,35]. However, in the last few years the existence of alternate routes, besides the one including EIN2, is further supported by different experimental results (Figure 1; [7,30,32,41]).

In relation to the Fe nutrition of dicot (Strategy I) plants, it has been shown that some Fe deficiency responses are impaired in the Arabidopsis ET insensitive mutant *ein2-1*. For example, this mutant does not develop subapical root hairs either under Fe deficiency or upon ACC treatment, while the wild-type Columbia does Figure 10 [13]. Moreover, ferric reductase activity (FRA) and *FRO2* and *IRT1* expression are induced in the *ein2-1* mutant both under Fe deficiency and upon ET treatments [27,42,53]. However, *FRO2* and *IRT1* expression attained much lower levels, and was more delayed, in this mutant than in the wild-type Columbia [53].

The results obtained in this work show that both *Arabidopsis* ET insensitive mutants, *ein2-1* and *ein2-5*, are able to respond to Fe deficiency by inducing some physiological responses, such as FRA enhancement and *FRO2*, *IRT1* and *FIT* upregulation (Figure 2 and Figure 3), which agrees with previous results obtained with the *ein2-1* mutant [27,42]. However, none of the mutants induced subapical root hairs either under Fe deficiency or upon ACC/GSNO treatments (Figure 10), which also matches with previous results obtained with both mutants upon ACC treatment, under Fe deficiency and under other nutrient deficiencies [13,24,27,49,54]. At first, these results suggest that physiological and morphological responses may be regulated through different ET signaling pathways, as previously suggested [13,27,42]. In relation to the physiological responses, the similar behavior of both mutants in their response to Co for FRA enhancement (Figure 3) but their different behavior in response to Co for *FRO2* and *IRT1* expression (Co, at the concentrations used, inhibits their upregulation in *ein2-1* but not in *ein2-5*; Figure 2) suggests that the *ein2-1* mutation has distinct consequences than the *ein2-5* one. Furthermore, in the *ein2.5* mutant, Co did not inhibit *FRO2* expression (responsible for FRA; Figure 2d) but inhibited FRA enhancement (Figure 3b), which would suggest a posttranscriptional regulation of *FRO2*, as previously proposed by Connolly et al. [55]. These results also suggest a possible implication for ET in such a process. It should be noted that the addition of Co, at 50 µM final concentration, to Fe-deficient wild-type Columbia plants drastically inhibited both FRA enhancement and *FIT*, *FRO2* and *IRT1* upregulation [27,46]. However, Co, even at 100 µM final concentration, did not inhibit either *FRO2* or *IRT1* expression in the *ein2-5* mutant (Figure 2d,e).

To further analyze the differences between the *ein2-1* and *ein2-5* mutations, we then compared the behavior of both mutants in response to ACC (ET precursor) and GSNO (NO donor) treatments. As shown in Figure 4 and Figure 5, both mutants induced all the Fe deficiency physiological responses studied (FRA and *FRO2*, *IRT1* and *FIT* expression) in response to ACC treatments, as occurred in wild-type Columbia plants [27,46]. These results agree with previous ones obtained with *ein2-1* mutant plants treated with either ACC or ET itself [27,42]. However, the intensity of *FRO2* and *IRT1* upregulation, and FRA enhancement, in response to ACC treatment was lower in the *ein2-5* mutant than in the *ein2-1* mutant, again suggesting that both mutations have different consequences.

Since NO has also been involved in the activation of Fe deficiency physiological responses [14,15,26,27], we studied the effect of GSNO application on the induction of these responses in both mutants. Similarly to ACC, GSNO also induced all the Fe deficiency physiological responses studied (FRA and *FRO2*, *IRT1* and *FIT* expression) in the *ein2-1* mutant (Figure 7 and Figure 8), as occurred in wild-type Columbia plants [26,27]. FRA was only induced in Fe-deficient plants (Figure 9), which further supports the existence of Fe-related repressive signals [46,51]. In contrast to the *ein2-1* mutant, GSNO had no positive effect on the *ein2-5* mutant (Figure 7 and Figure 8); only a slight FRA enhancement in Fe-deficient plants (Figure 9). This again shows that the *ein2-5* mutant is less responsive to the GSNO treatment, similarly to what occurs with ACC (see above paragraph).

In previous works, NO accumulation in the subapical region of roots either under Fe deficiency or upon ACC treatment has been found [14,24,26]. To test this possibility, both mutants, and also wild-type Columbia plants, were subjected to either Fe deficiency or Fe sufficiency with ACC addition. The *ein2-1* mutant accumulated NO under both kinds of treatments, as occurred in the wild-type Columbia, but the *ein2-5* mutant only accumulated NO under Fe deficiency but not upon ACC treatment (Figure 6). This again shows that this latter mutant is less responsive to ACC than the *ein2-1* mutant. In addition, the results indicate that EIN2 plays an important role in the accumulation of NO upon ACC treatment, but that perhaps this role could be overridden under Fe deficiency.

The results obtained in this work clearly support the participation of ET, through EIN2, in the regulation of the Fe acquisition genes *FRO2* and *IRT1.* Furthermore, the results do support the existence of additional routes for ET signaling besides the linear canonical one including EIN2 (Figure 1) [7,30,32,41]. Even more, since the less responsive mutant to ET (ACC) treatment, *ein2-5*, it still able to upregulate Fe acquisition genes under Fe deficiency (Figure 2), the existence of additional signaling routes, ET-independent, for the activation of the genes would be also possible (Figure 1). In this sense, it should be noted that ACC itself has been proposed as a signaling molecule [56] and other hormones, such as auxin, have also be involved in the upregulation of Fe acquisition genes (Figure 1) [17,22]. There are also hormones affected by Fe deficiency, such as jasmonic acid (JA), that could negatively interact with the ET signaling pathway. Kim et al. [43] showed that both *ein2-1* and *ein2-5* mutants become responsive to ET if JA levels or signaling are low. In accordance with these results, it should be noted that a role for JA in the suppression of Fe deficiency responses has been proposed [57,58]. Another possibility could be the existence of additional FIT-independent pathways to control *FRO2* and *IRT1* expression. In this way, Balparda et al. [45] have recently proposed that *FRO2* and *IRT1* expression could be directly controlled by the ERF1 TF (also associated with ET; see Introduction). The possibility exists that these alternate pathways (Figure 1) could be potentiated when the EIN2-dependent pathway is impaired.

## 4. Materials and Methods

### 4.1. Plant Materials, Growth Conditions and Treatments

Seeds of the *Arabidopsis thaliana* (L.) Heynh ecotype “Columbia” and the ET-insensitive mutants *ein2-1* and *ein2.5* were grown under controlled conditions as previously described [27]. Briefly, seeds were germinated in black peat and, when appropriate, seedlings were transferred to individual containers (of 70 mL volume) with complete nutrient solution continuously aerated. The nutrient solution without Fe had the following composition: macronutrients; 2 mM Ca(NO_3_)_2_, 0.75 mM K_2_SO_4_, 0.65 mM MgSO_4_, 0.5 mM KH_2_PO_4_; and micronutrients; 50 µM KCl, 10 µM H_3_BO_3_, 1 µM MnSO_4_, 0.5 µM CuSO_4_, 0.5 µM ZnSO_4_, 0.05 µM (NH_4_)_6_Mo_7_O_24_. Fe-EDDHA was added to the nutrient solution at different concentrations (10 or 40 µM Fe-EDDHA) depending on the experiments. Plants were grown in a growth chamber at 22 °C day/20 °C night, with relative humidity between 50% and 70%, and an 8 h photoperiod (to postpone flowering) at a photosynthetic irradiance of 300 µmoL m^−2^ s^−1^ provided by fluorescent tubes (Sylvania Cool White VHO).

The treatments imposed were: **Fe40**: nutrient solution with 40 µM Fe-EDDHA; **Fe10**: nutrient solution with 10 µM Fe-EDDHA; **Fe10 + ACC**: Fe10 treatment with ACC addition, at 0, 1 or 5 µM final concentration, during the last 24 h; **Fe10 + GSNO**: Fe10 treatment with GSNO addition, at 100 µM final concentration, during the last 24 h; –**Fe**: nutrient solution without Fe (24 h or 48 h depending on the experiments); –**Fe + GSNO**: –Fe treatment 24 h with GSNO addition, at 100 or 500 µM final concentration; –**Fe + Co**: –Fe treatment 48 h with CoSO_4_ addition, at 0, 50, 75 or 100 µM final concentration, during the last 24 h. In our experimental conditions, plants appreciably induced Fe deficiency responses after 2–3 days of Fe deficiency. Consequently, in the -Fe + GSNO treatment, the idea was to accelerate the induction of the responses with the GSNO treatment, while in the -Fe + Co treatments, the idea was to inhibit the induction of the responses by the Co treatment. Stock solution of GSNO was prepared as previously described [27]. After treatments, root ferric reductase activity (FRA) was determined as described in the next section. Finally, the roots were collected and kept at −80 °C to later analyze gene expression. Each treatment consists of six biological replications. In some cases, “Columbia”, *ein2-1* and *ein2.5* plants growing in nutrient solution with 10 µM Fe-EDDHA were treated for 24 h with ACC (1 or 10 µM final concentration) or GSNO (100 or 500 µM final concentration). After that, roots were excised and stained with toluidine blue (0.05% *w/v*) and pictures were taken by using a stereomicroscope.

### 4.2. Ferric Reductase Activity Determination

FRA was determined as previously described [46]. Briefly, intact plants were pretreated for 30 min in plastic vessels with 50 mL of a nutrient solution without micronutrients, pH 5.5, and then placed into 20 mL of a Fe (III) reduction assay solution for 1 h. This assay solution consisted of nutrient solution without micronutrients, 100 µM Fe (III)-EDTA and 300 µM Ferrozine, pH 5.0 (adjusted with 0.1N KOH). The environmental conditions during the measurement of Fe (III) reduction were the same as the growth conditions described above. FRA was determined spectrophotometrically by measuring the absorbance (562 nm) of the Fe (II)-Ferrozine complex and by using an extinction coefficient of 29,800 M^−1^ cm^−1^. After the reduction assay, roots were excised and weighed, and the results were expressed on a root fresh weight basis. The values represent the mean ± SE of six replicates.

### 4.3. Real-Time PCR Analysis

Real-time PCR analysis was carried out as previously described [53]. Briefly, roots were ground to a fine powder with a mortar and pestle in liquid nitrogen. Total RNA was extracted using the Tri Reagent solution (Molecular Research Center, Inc., Cincinnati, OH, USA) according to the manufacturer’s instructions. cDNA synthesis was performed by using M-MLV reverse transcriptase (Promega, Madison, WI, USA) from 3 µg of DNase-treated root RNA as the template and random hexamers as the primers. The gene expression study by qRT-PCR was performed in a qRT-PCR Bio-Rad CFX connect thermal cycler and the following amplification profile: initial denaturation and polymerase activation (95 °C for 3 min), amplification and quantification repeated 40 times (90 °C for 10 s, 57 °C for 15 s and 72 °C for 30 s), and a final melting curve stage of 65 °C to 95 °C with increment of 0.5 °C for 5 s to ensure the absence of primer dimer or non-specific amplification products. PCR reactions were set up in 20 µL of SYBR Green Bio-RAD PCR Master Mix, following the manufacturer’s instructions. Controls containing water instead of cDNA were included to check for contamination in the reaction components. Primer pairs designed by García et al. [50] were used to amplify *FRO2*, *IRT1*, and *FIT* cDNA. Standard dilution curves were performed for each primer pair to confirm appropriate efficiency of amplification (E = 100 ± 10%). Constitutively expressed *SAND1* and *YLS8* genes, which do not respond to changes in the Fe conditions [59], were used as reference genes to normalize qRT-PCR results. The relative expression levels were calculated from the threshold cycles (Ct) values and the primer efficiencies by the Pfaffl method [60]. Each PCR analysis was conducted on three biological replicates and each PCR reaction was repeated twice.

### 4.4. NO Localization

Nitric oxide (NO) was imaged using 4,5-diaminofluorescein diacetate (DAF-2 DA) and epifluorescence microscopy as previously described [26]. Roots were loaded with 5 µM DAF-2 DA in 10 mM HEPES/NaOH pH 7.5 buffer for 1 h, washed 3 times in fresh buffer and analyzed microscopically (Leika DMRB; excitation 488 nm, emission 495–575 nm).

### 4.5. Statistical Analyses

All experiments were repeated at least twice and representative results are presented. The values of qRT-PCR represent the mean ± SE of three independent biological replicates. The values of FRA represent the mean ± SE of six replicates. Depending on the experiment, different tests were used. When comparing different treatments with a control (Figure 4 and Figure 5), *, ** or *** indicate significant differences (*p* < 0.05, *p* < 0.01 or *p* < 0.001) among treatments using one-way analysis of variance (ANOVA) followed by a Dunnett’s test. Different letters (Figure 2 and Figure 3) indicate significant differences (*p* < 0.05) among treatments using one-way analysis of variance (ANOVA) followed by a Tukey multiple range test. In the GSNO experiments (Figure 7, Figure 8 and Figure 9), *, ** or *** indicate significant differences (*p* < 0.05, *p* < 0.01 or *p* < 0.001) between the two treatments according to the Student’s *t* test.

## 5. Conclusions

The results presented in this work show that *Arabidopsis* ethylene insensitive *ein2* mutants are still able to respond to ET (ACC) treatments for the upregulation of Fe deficiency physiological responses, which agrees with previous results [27,42]. In the same way, they are able to respond to NO treatments. However, the *ein2-5* mutant (altered in the Nramp-domain of EIN2) and the *ein2-1* mutant (altered in the CEND portion of EIN2) differ in their response to ACC (ET precursor) and GSNO (NO donor) treatments, with *ein2-5* being less responsive to both of them. This suggests that ET and NO are closely interrelated through EIN2. However, the reasons for the differences between *ein2-1* and *ein2-5* are not yet known.

The results obtained clearly support the participation of ET and NO, through EIN2, in the regulation of the Fe acquisition genes *FRO2* and *IRT1.* In the same way, the results also support the implication of ET and NO, through EIN2, in the development of subapical root hairs. Furthermore, the results do support the existence of additional routes for ET signaling besides the linear canonical one including EIN2.

Since the less responsive mutant to ET (ACC) and NO (GSNO) treatments, *ein2-5*, is still able to upregulate Fe acquisition genes under Fe deficiency, this could suggest the existence of additional signaling routes, ET-independent, for the activation of these genes.

Finally, the results obtained show that cobalt (ET inhibitor) inhibits the enhanced ferric reductase activity provoked by Fe deficiency in the *ein2-5* mutant, while the *FRO2* gene, associated with this activity, is not inhibited. These results further support the posttranscriptional regulation of *FRO2*, suggesting a possible implication of ET in such a process.

## Figures and Tables

**Figure 1 plants-10-00262-f001:**
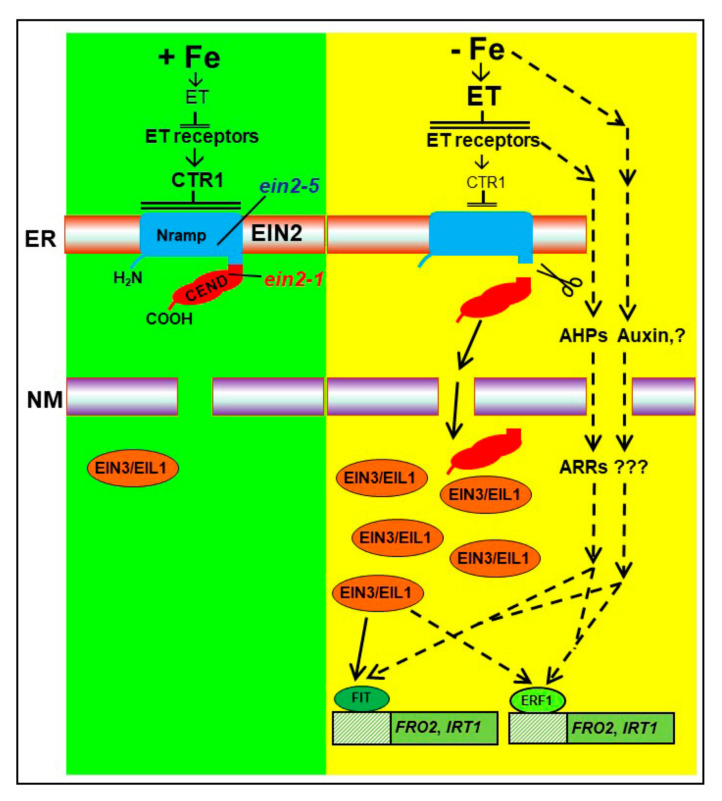
Model for the role of ethylene, through EIN2, on the upregulation of the Fe acquisition genes *FRO2* and *IRT1* in Arabidopsis. Fe deficiency causes enhanced ET production in roots. In the presence of ET, the CEND portion of the EIN2 protein is cleaved and shuttled into the nucleus where it increases the transcriptional activity of the EIN3/EIL1 TFs (see Introduction for more details). The accumulation of these latter TFs trigger ET responses, such as enhanced *FIT* transcription and, consequently, *FRO2* and *IRT1* upregulation. The transcription of *FIT*, *FRO2* and *IRT1* could also be triggered through alternate pathways, such as the “non canonical” ET-signaling route involving AHPs and ARRs or the ones involving other hormones, such as auxin (dashed lines). In addition to *FIT*, the different routes could also upregulate *FRO2* and *IRT1* through other TFs, such as ERF1. Abbreviations: AHPs, Arabidopsis Histidine-containing Phosphotransmitters; ARRs, Arabidopsis Response Regulators; ER, Endoplasmic Reticulum; ERF1, Ethylene Response Factor1; ET, Ethylene; NM, Nuclear Membrane; TFs, Transcription Factors. Based on [7,17,32,33,34,35,36,44,45].

**Figure 2 plants-10-00262-f002:**
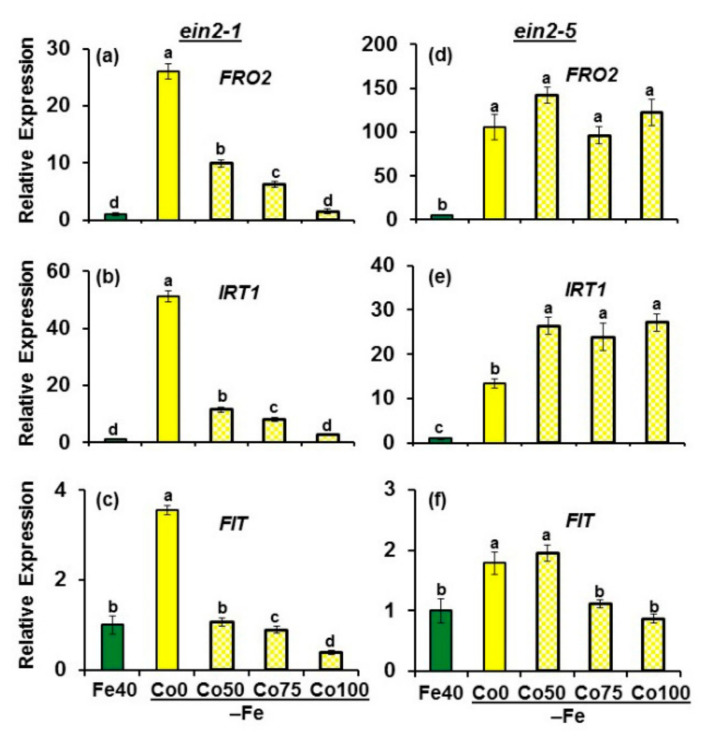
Effect of Fe deficiency and cobalt (Co; an ET synthesis inhibitor) on *FRO2*, *IRT1* and *FIT* expression in roots of the Arabidopsis ET insensitive mutants *ein2-1* (**a**–**c**) and *ein2-5* (**d**–**f**). Plants were grown in complete nutrient solution. When appropriate, some of them were transferred during 48 h to complete nutrient solution with 40 µM Fe-N,N’-ethylenebis[2-(2-hydroxyphenyl)-glycine (Fe-EDDHA) (Fe40) or without Fe (–Fe). Co, at different final concentrations (0, 50, 75 or 100 µM), was added to the nutrient solution without Fe during the last 24 h. After treatments, roots were collected and kept at −80 °C for subsequent analysis of mRNA levels. *FRO2*, *IRT1* and *FIT* expression were determined. Relative expression was calculated in relation to the Fe40 treatment. Data represent the mean of 3 independent biological replicates ± S.E. Bars with different letters indicate significant differences (*p* < 0.05) according to the Tukey’s test.

**Figure 3 plants-10-00262-f003:**
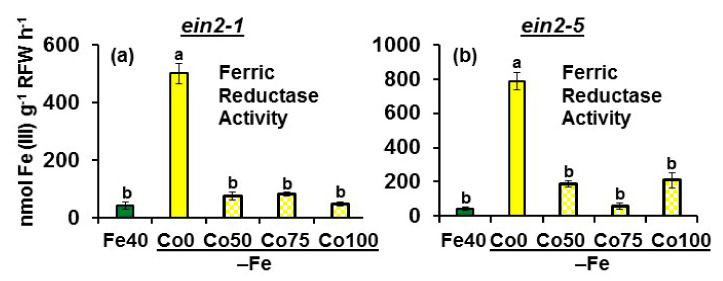
Effect of Fe deficiency and cobalt (Co; an ET synthesis inhibitor) on the ferric reductase activity (FRA) of the Arabidopsis ET insensitive *ein2-1* (**a**) and *ein2-5* (**b**) plants. Treatments as in Figure 2. After treatments, FRA was determined. Data represent the mean of 6 replicates ± S.E. Bars with different letters indicate significant differences (*p* < 0.05) according to the Tukey’s test.

**Figure 4 plants-10-00262-f004:**
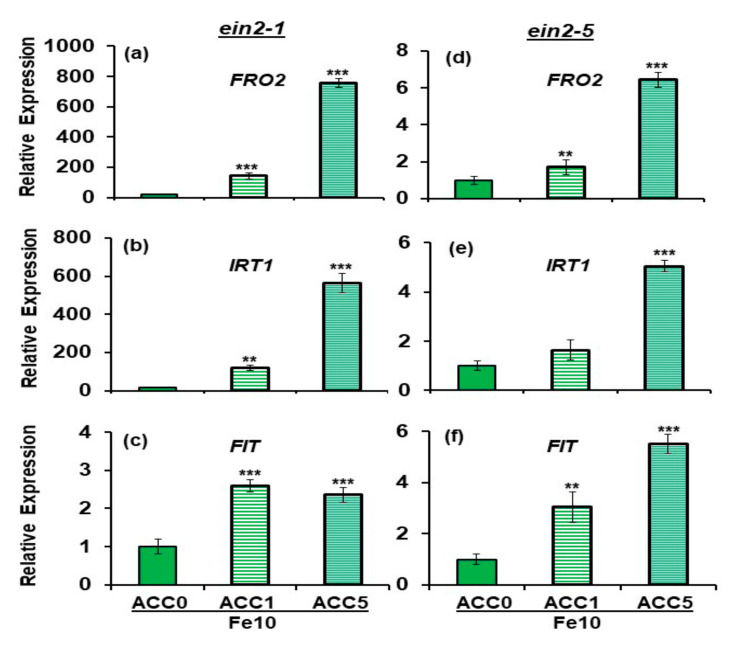
Effect of 1-aminocyclopropane-1-carboxylate (ACC) (ET precursor) on *FRO2*, *IRT1* and *FIT* expression in roots of the Fe-sufficient Arabidopsis ET insensitive mutants *ein2-1* (**a**–**c**) and *ein2-5* (**d**–**f**). Plants were grown in complete nutrient solution with 10 µM Fe-EDDHA (Fe10). ACC, at different final concentrations (0, 1 or 5 µM), was added to the nutrient solution during the last 24 h. After treatments, roots were collected and kept at −80 °C for subsequent analysis of mRNA levels. *FRO2*, *IRT1* and *FIT* expression were determined. Relative expression was calculated in relation to the Fe10 treatment. Data represent the mean of 3 independent biological replicates ± S.E. Bars with ** or *** indicate significant differences (*p* < 0.01 or *p* < 0.001) in relation to the Fe10 treatment according to the Dunnett’s test.

**Figure 5 plants-10-00262-f005:**
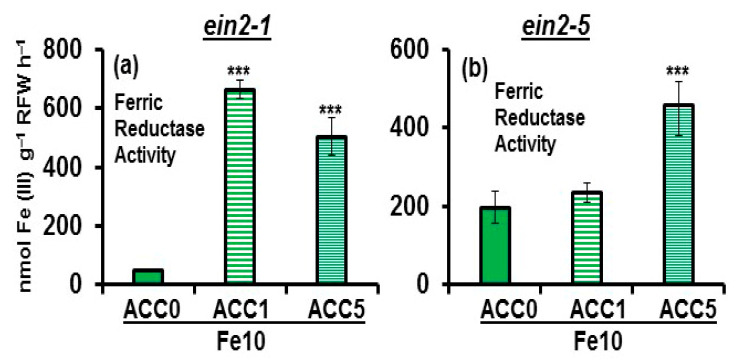
Effect of ACC (ET precursor) on the ferric reductase activity (FRA) of Fe-sufficient Arabidopsis ET insensitive *ein2-1* (**a**) and *ein2-5* (**b**) plants. Treatments as in Figure 4. After treatments, FRA was determined. Data represent the mean of 6 replicates ± S.E. Bars with *** indicate significant differences (*p* < 0.001) in relation to the Fe10 treatment according to the Dunnett’s test.

**Figure 6 plants-10-00262-f006:**
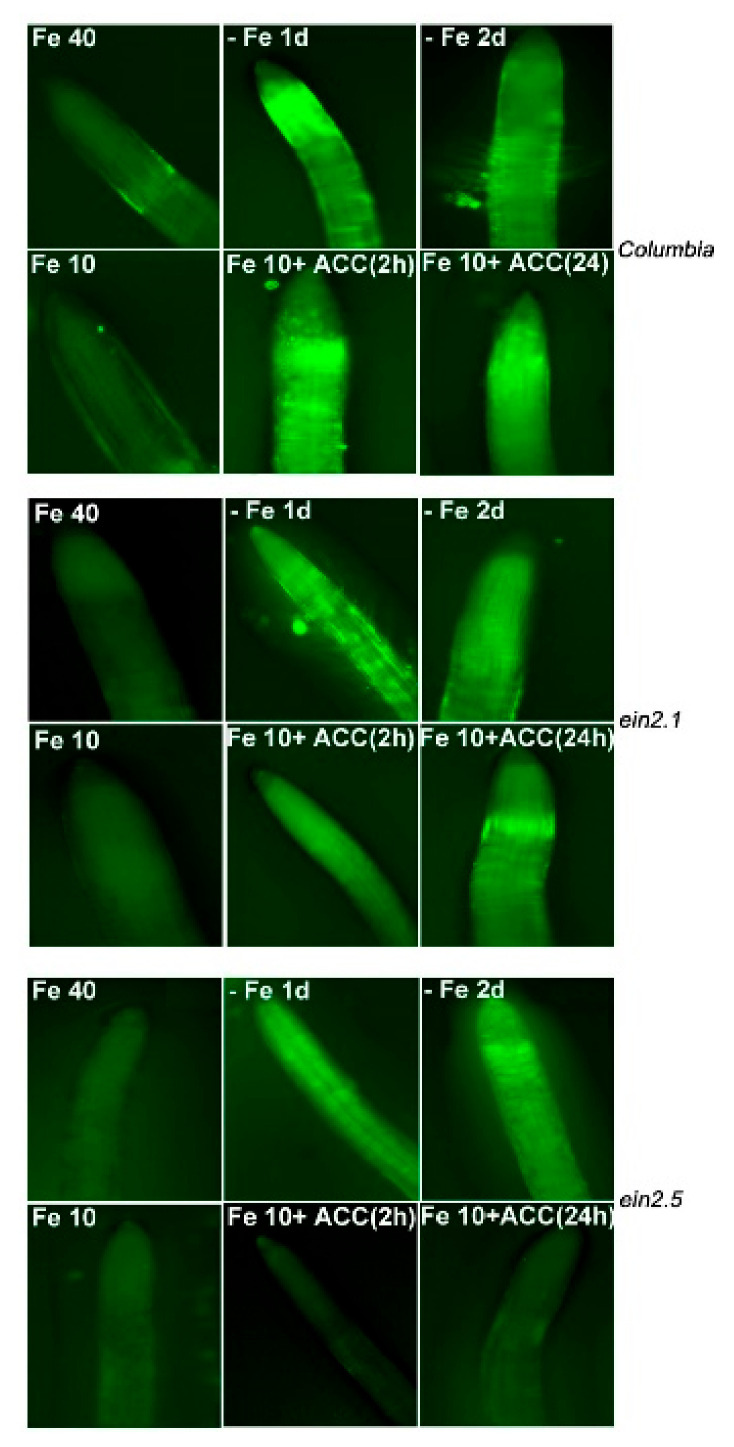
Effect of Fe deficiency and ACC on nitric oxide (NO) accumulation by roots of Arabidopsis wild-type Columbia and the ET insensitive mutants *ein2-1* and *ein2-5*. Plants were grown in complete nutrient solution. When appropriate, some of them were transferred to complete nutrient solution with 40 µM Fe-EDDHA (Fe40), with 10 µM Fe-EDDHA (Fe10) or without Fe, either during 1 day (–Fe 1d) or 2 days (–Fe 2d). ACC, at 1 µM final concentration, was added to some of the plants in the Fe10 treatment during either 2 h [Fe10+ACC (2 h)] or 24 h [Fe10 + ACC (24 h)]. NO was visualized with the NO-sensitive fluorescent dye DAF-2 DA. Notice the localization of NO accumulation, induced by either ACC treatment or Fe deficiency, on the subapical region of the roots.

**Figure 7 plants-10-00262-f007:**
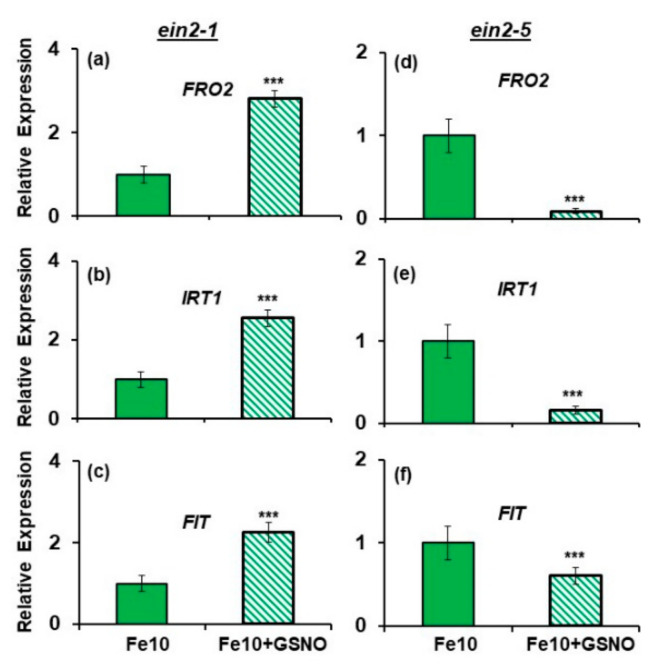
Effect of GSNO (NO donor) on *FRO2*, *IRT1* and *FIT* expression in roots of the Fe-sufficient Arabidopsis ET insensitive mutants *ein2-1* (**a**–**c**) and *ein2-5* (**d**–**f**). Plants were grown in complete nutrient solution with 10 µM Fe-EDDHA (Fe10). GSNO, at 100 µM final concentration, was added to the nutrient solution of half of the plants during the last 24 h. After treatments, roots were collected and kept at −80 °C for subsequent analysis of mRNA levels. *FRO2*, *IRT1* and *FIT* expression were determined. Relative expression was calculated in relation to the Fe10 treatment. Data represent the mean of 3 independent biological replicates ± S.E. Bars with *** indicate significant differences (*p* < 0.001) in relation to the Fe10 treatment according to the Student’s test *t*.

**Figure 8 plants-10-00262-f008:**
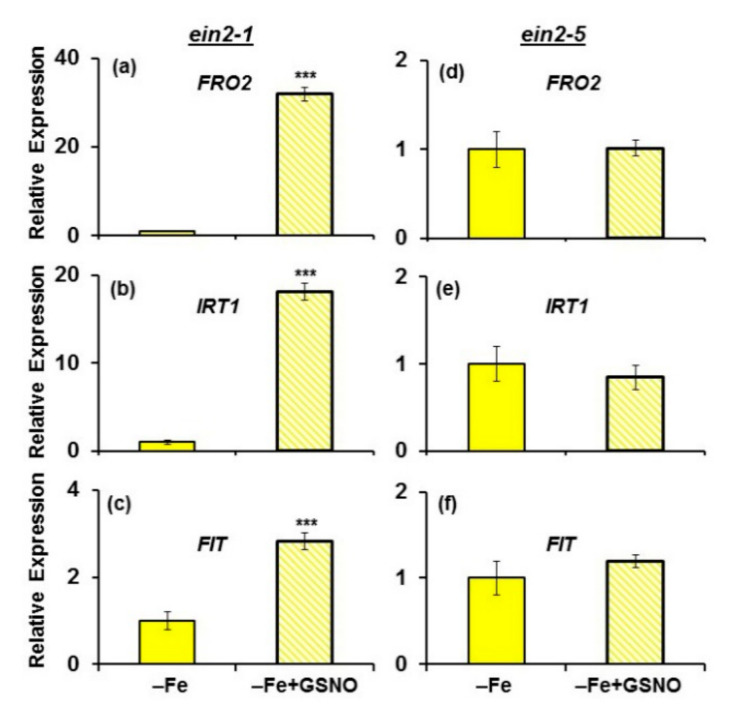
Effect of GSNO (NO donor) on *FRO2*, *IRT1* and *FIT* expression in roots of the Fe-deficient Arabidopsis ET insensitive mutants *ein2-1* (**a**–**c**) and *ein2-5* (**d**–**f**). Plants were grown in complete nutrient solution. When appropriate, they were transferred during 24 h to nutrient solution without Fe (–Fe). GSNO, at 100 µM final concentration, was added to the nutrient solution of half of the plants. Determinations as in Figure 7. Bars with *** indicate significant differences (*p* < 0.001) in relation to the –Fe treatment according to the Student’s test *t*.

**Figure 9 plants-10-00262-f009:**
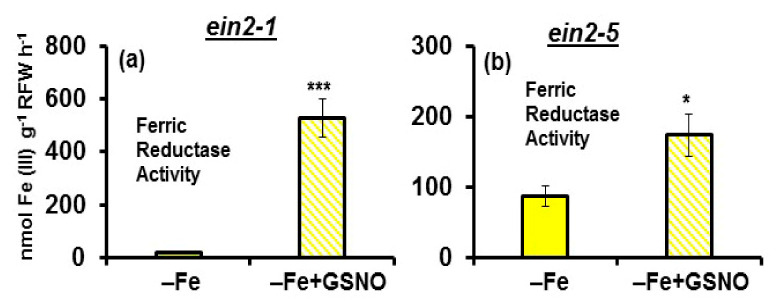
Effect of GSNO (NO donor) on the ferric reductase activity (FRA) of Fe-deficient Arabidopsis ET insensitive *ein2-1* (**a**) and *ein2-5* (**b**) plants. Plants were grown in complete nutrient solution. When appropriate, they were transferred during 24 h to nutrient solution without Fe (–Fe). GSNO, at 100 µM final concentration, was added to the nutrient solution of half of the plants. After treatments, FRA was determined. Data represent the mean of 6 replicates ± S.E. Bars with * or *** indicate significant differences (*p* < 0.05 or *p* < 0.001) in relation to the –Fe treatment according to the Student’s test *t*.

**Figure 10 plants-10-00262-f010:**
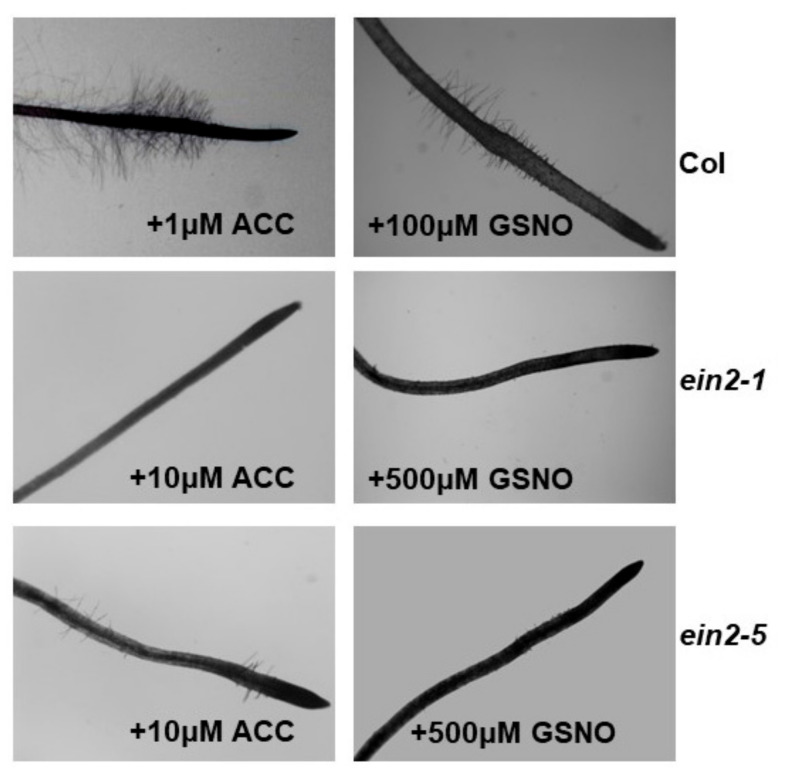
Effect of ACC and GSNO on the development of subapical root hairs by Fe-sufficient Arabidopsis wild-type Columbia, ET insensitive *ein2-1* and *ein2-5* plants. Plants were grown in complete nutrient solution with 10 µM Fe-EDDHA. Some of them were treated for 24 h with ACC (1 or 10 µM final concentration) or GSNO (100 or 500 µM final concentration). After that, roots were excised and stained with toluidine blue (0.05% *w/v*). Pictures were taken by using a stereomicroscope.

## Data Availability

All the data included in this article are publicly available.

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
