# Peer review of "Comparative Study of Several Fe Deficiency Responses in the Arabidopsis thaliana Ethylene Insensitive Mutants ein2-1 and ein2-5"

_plants, 2021, doi:10.3390/plants10020262_

Round 1
Reviewer 1 Report
Dear Editor,
In the manuscript with the title “Comparative study of several Fe deficiency responses in the Arabidopsis thaliana ethylene insensitive mutants ein2-1 and ein2-5” Macarena et al try to elucidate the role of ethylene pathway to Fe-deficiency by using two different mutant alleles of EIN2, ein2-1 located in the CEND domain and ein2-5 located at the Nramp domain of EIN2 protein. Based on previous work they studied through qPCR the expression of FRA, FIT, FRO2 and IRT1 genes implicated in Fe deficiency response in the mutant backgrounds of ein2-1 and ein2-5.
Macerena and co-authors compared the transcriptional profiles of the above genes as well as the activity of ferric reductase in ein2-1 and ein2-5 mutant plants grown under different concentrations of Co and in the presence of ACC and GSNO. In the presented results it is obvious that the transcriptional regulation of the Fe-deficiency related genes is different in ein2-1 and ein2-5 in the experimental conditions the authors used.
My main concern is whether the observed differences in the expression of FRA, FIT, FRO2 and IRT1 or the different ferric reductase activity between ein2-1 and ein2-5 mutants is extrapolated or reflected in the development of the root of these mutants or these differences are masked.
For this reason, I suggest that physiological measurements concerning Fe-deficiency induced morphological responses in root development, like primary root length or root hair development or root branching or lateral root development, is necessary to be applied in wild-type and ein2-1 and ein2-5 mutants to support the findings of the different expression levels of Fe-deficiency markers. Comparison of the response of ein2-1 and ein2-5 in the different treatments will further support the different responses of ein2-1 and ein2-5 mutants. In detail
- In Figure 2 physiological assays concerning root development should be performed for the treatments used for qPCR analysis.
- In Figure 4 physiological assays for root development should be performed for the treatments used for qPCR analysis.
- In Figure 7 physiological assays for root development should be included for treatments used for qPCR analysis
- In Figure 8 . The same as previous.

Reviewer 2 Report
The authors study the gene interaction by ethylen and ON in Arabidopsis under Fe deficiency. Introduction and methodology are adequate and the experimental design in accordance to the objectives. The discussion of the complex results is sound giving new approxing conclusions.
The authors adress a new original aproach to the complex molecular interactions in Arabidopsis ethylene insensitive ein2 mutants under Fe deficiency. The basic points are Co, ethylene and NO. This results gives clues for future work in relation to the differential action of ethylene and NO.
In the text the genes should be expressed in cursive script.
Round 2
Reviewer 1 Report
Dear Editor,
The authors have addressed sufficiently my concerns about the previous version of the manuscript